# Considerations for Purchasing Drug Checking Technologies: Perspectives from Toronto’s Drug Checking Service

**DOI:** 10.3390/ijerph20156486

**Published:** 2023-07-31

**Authors:** Hayley Thompson, Karen McDonald

**Affiliations:** Toronto’s Drug Checking Service, Centre on Drug Policy Evaluation, St. Michael’s Hospital, Unity Health Toronto, 30 Bond Street, Toronto, ON M5B 1W8, Canada; kn.mcdonald@utoronto.ca

**Keywords:** drug checking, drug checking technologies, harm reduction

## Abstract

With the unregulated drug supply—particularly the unregulated opioid supply—becoming increasingly more toxic, more contaminated, and less predictable, drug checking has emerged as an essential public health service: informing individuals who use drugs, as well as those who care and advocate for them, in real-time. For those looking to offer drug checking services in community settings, choosing a technology can be an arduous task. With very little regulatory oversight of drug checking technologies, it can be difficult for organizations that specialize in harm reduction to ascertain what questions to ask drug checking technology vendors to ensure they invest in a technology that best suits the needs of their community. Looking to help those that lack drug checking and technical expertise, Toronto’s Drug Checking Service has compiled a list of questions to equip organizations to make informed decisions when it comes to purchasing drug checking technologies. Having developed and operated a drug checking service since 2018, Toronto’s Drug Checking Service is uniquely positioned to share its expertise and insights.

## 1. Introduction

The need for, and use of, drug checking or “street drug analysis” to inform dosing and drug use decisions, as well as to contextualize adverse health events experienced by people who use drugs, has been documented as far back as the 1970s in both the United States and the Netherlands [1,2]. As Maghsoudi et al. note in their 2022 systematic review, the proliferation of drug checking services in Europe occurred in the 1990s as a harm reduction measure in festival or nightlife settings [3]. In recent years, in response to exponential rates of opioid-related morbidity and mortality in North America, in particular, drug checking has become part of a suite of services offered in community settings to those at highest risk of drug poisoning, enabling them to make informed drug use decisions by providing detailed information on the composition of their drugs [3].

With more than 34,400 apparent opioid toxicity deaths between January 2016 and September 2022, Canada is in the midst of a national public health crisis [4,5]. In response to the alarming number of adverse health events and mortality caused by the toxic drug supply, the Canadian government funded three drug checking pilot programs in 2018—two in British Columbia and one in Ontario—with hopes of assessing the impact of drug checking on unintentional drug poisoning-related risks.

Launched in October of 2019, Toronto’s Drug Checking Service remained the only robust and long-term drug checking program in the province of Ontario until late 2022. Since the service’s launch, over 50 community-based organizations and regional health authorities have contacted Toronto’s Drug Checking Service to learn about the operations of the service, to discuss potential partnership opportunities, or to understand the resources required to establish local services.

In Canada, drug checking technologies are not assessed to determine their safety, effectiveness, or quality before being authorized for sale—a requirement for medical devices, which are approved by Health Canada, the federal institution responsible for overseeing the regulation of drugs and health-related products [6]. With so many organizations reaching out to Toronto’s Drug Checking Service looking to initiate local drug checking programs and invest in technologies, as well as the emergence of new technologies, understanding the potential limitations of drug checking technologies and the information they are able to provide is vital.

## 2. Overview of Toronto’s Drug Checking Service

Toronto’s Drug Checking Service is a free and anonymous public health service, currently available at five community health agencies in downtown Toronto, Canada, where supervised consumption services are also offered. Accepted samples include a small (10 mg) sample of a substance or drug equipment (e.g., cookers or filters) after it has been used. Results are available within a business day or two and are communicated to service users by harm reduction staff in person, by phone, or by email, along with tailored strategies to reduce harm and referrals to drug-related, health, and social services [7].

Toronto’s Drug Checking Service currently operates using an “offsite” drug checking model, where samples are transported to nearby clinical laboratories (the Centre for Addiction and Mental Health or St. Michael’s Hospital) and analyzed using gas chromatography (GC)- or liquid chromatography (LC)-mass spectrometry (MS). During the program planning phase, it was decided that Toronto’s Drug Checking Service would leverage already existing instruments, human resources, and community-based services to optimize program funding. At the time of planning, it was not known how critical laboratory-based technologies (like GC- and LC-MS) are when detecting rare and novel drugs, as well as drugs found in very trace amounts. There were also limited data on the effectiveness of using these technologies to validate results from other, often portable, drug checking technologies. Other advantages of utilizing GC- and LC-MS technologies include that they are able to break apart complex drug mixtures and differentiate between drugs that are chemically similar but may produce very different effects (e.g., fentanyl and carfentanil). Lastly, these technologies can be used to provide precise quantity-based information on exactly how much of a particular drug is present in a sample. Having such sophisticated technologies for routine analysis has proven to be very advantageous for drug market monitoring in the City of Toronto, and for others in Ontario who use the service’s findings to contextualize the experiences people who use drugs report in other jurisdictions. The disadvantages of using GC- and LC-MS for routine analysis are that they are costly and turnaround times are longer than those of portable, point-of-care, or “onsite” drug checking technologies, which means individuals accessing the service must wait a business day or two for drug checking results. Despite longer turnaround times, 99% of service users surveyed by Toronto’s Drug Checking Service have found the program useful and 84% plan to use the service again, speaking to the value of the service amongst the community of people who use drugs in downtown Toronto [8].

Every other week Toronto’s Drug Checking Service aggregates results from samples checked and publicly disseminates unregulated drug supply trends in an effort to keep people who use drugs, harm reduction workers, clinicians, researchers, policy makers, and others apprised of what is circulating in the city’s unregulated drug supply in real-time [7].

## 3. Discussion

From October 2019 to November 2022, the service checked 7644 drug samples from Toronto’s unregulated supply, using GC- or LC-MS: approximately 50% (3765) were expected to be (i.e., obtained or bought as) opioids. The remaining 50% (3879) were expected to be a variety of stimulants, depressants, and psychedelics. Having used GC- and LC-MS for analysis of all 7644 samples checked, Toronto’s Drug Checking Service feels strongly that technologies with equivalent GC-, LC-, or paper spray- (PS), MS capabilities are currently the only known available technologies capable of providing service users with the level of detail required to make fully informed opioid use decisions. When it comes to other drugs in Toronto, such as most stimulants, psychedelics, and depressants, it is the opinion of Toronto’s Drug Checking Service that other technologies, such as Fourier-transform infrared spectroscopy (FTIR) paired with test strips, should be able to provide adequate drug checking results (this may not be the case for all jurisdictions, especially those where non-opioid drugs are confirmed as or suspected of being highly contaminated). At present, there is no perfect drug checking technology—each has tradeoffs with respect to cost, the quality of results it can provide, and timeliness of results. The goal in sharing the information contained within this publication is not to dissuade organizations from purchasing any particular technology, but to encourage them to think critically when deciding what technology is best given the local context.

Toronto’s Drug Checking Service wishes to impart emphatically to all those implementing drug checking services that drug checking is not easy. From the perspective of Toronto’s Drug Checking Service, those offering drug checking services should:Have a keen understanding of what information is most important to potential service users and the community in order to acquire the most suitable technology (e.g., how important are immediate results? How important is the identification and reporting of all drugs found in a sample, including those in trace amounts? How important is the identification and reporting of non-drug fillers? Are service users willing to give up a small amount of their drug for a destructive technology?).Deeply understand the limitations of the technology they are purchasing and communicate those limitations to service users in plain language (e.g., which compounds the technology can and cannot identify, as well as the technology’s limit of detection (i.e., the smallest amount of a compound that can be detected with confidence)).

Organizations such as the British Columbia Centre on Substance Use [9], the Drug Resource and Education Project [10], and the Trans European Drug Information (TEDI) [11] project have created thorough and well-informed resources that highlight the benefits, limitations, costs, and other considerations of many currently available drug checking technologies in an effort to guide organizations to the technology that would best fit their needs. Building on this foundation, Toronto’s Drug Checking Service has identified a series of questions which organizations purchasing drug checking technologies may wish to ask vendors.

### 3.1. Company-Related Questions

Describe your organization’s mission, motivation, management team, legal structure and ownership, and revenue model.What experience does your organization have in the field of harm reduction, if any?How does your organization give back to the community of people who use drugs?

### 3.2. Technology-Related Questions

Describe how your technology works in lay terms.How has your technology been validated? Describe validation using reference standards (i.e., pharmaceutical grade known compounds in known amounts), as well as drugs from the unregulated supply, if applicable. Provide reports, peer-reviewed publications.*Provide detailed limitations for your technology.Has your technology been piloted in the community? If so, how and where, and could you provide a community contact we could connect with to learn about their experience?Is your technology new or does it build upon a technology already used for drug checking?What qualifications or training are required of those that conduct drug checks using your technology?How much physical space does your technology require? Is your technology portable? How durable is your technology (i.e., could it be used outdoors or in a vehicle)?

* As previously mentioned in this paper, drug checking technologies are considered consumer products in Canada and are not assessed by Health Canada to determine their safety, effectiveness, or quality before being authorized for sale in Canada (something that is a requirement for medical devices) [6]. For this reason, it is critically important that claims made about what a drug checking technology can do (specifically, which compounds it can detect) are backed by concrete evidence.

### 3.3. Cost and Maintenance-Related Questions

How much does your technology cost? What are upfront and ongoing costs related to subscriptions and supplies?How is your technology serviced? What are anticipated service and maintenance costs?How do we access instrument support? How long do we have to wait for instrument support? Are there costs associated with accessing instrument support?

### 3.4. Sample-Related Questions

What sample types can be checked using your technology? E.g., substances (powder, crystals, rocks, pills, blotter, liquid), residue on used drug equipment.Are samples checked in raw form or are they diluted? If diluted, with what?What expected drugs can be checked using your technology?Does your technology destroy the sample that is checked?

### 3.5. Results-Related Questions

How long are turnaround times for results?Which drugs can your instrument detect?Can your instrument detect non-drug fillers and other types of compounds? If so, which ones?How well does your instrument differentiate between drugs that have very similar chemical structures? I.e., can your instrument differentiate fentanyl-related drugs such as fentanyl, fluorofentanyl, and carfentanil? Or benzodiazepine-related drugs such as etizolam, flualprazolam, and bromazolam?Does your technology report information about how much of a compound is found in a checked sample (i.e., quantified results)? If so, within what range of precision (i.e., how accurately)?What is your technology’s limit of detection (i.e., the smallest amount of a compound that can be detected with confidence)? The limit of detection for a Fourier-transform infrared spectrometer (FTIR), which is currently the most used onsite drug checking technology for opioid overdose prevention in North America is 5% [10]. This means substances present at levels under 5% are likely to be missed by the instrument. For this reason, FTIR is paired with test strips, which are more likely to pick up certain drugs in trace amounts. Emerging drug checking technologies that prioritize opioid overdose prevention and claim to be improvements to existing onsite technologies should therefore have a limit of detection less than 5%.How often are “new” compounds added to your technology’s database or library of drugs it can detect?

### 3.6. Data-Related Questions

What data, if any, does your technology collect from service users? How is data stored? Where is the data stored? What does your organization do with the data? Are we free to do what we want with the data?Do you plan to share your drug sample analysis data with existing networks of publicly funded drug market monitoring systems for public dissemination?

### 3.7. Partnership-Related Questions

What benefits do community partners receive (e.g., free or discounted instruments)?

A PDF version of *Onsite drug checking technology purchase and partnership considerations* resource is available in the Appendix A section of this publication [12].

The above questions, taken from Toronto’s Drug Checking Service’s *Onsite drug checking technology purchase and partnership considerations* resource [12], were first drafted in April of 2022 in response to requests for advice and consultation from community-based organizations that were being marketed to by for-profit drug checking vendors. Once drafted, these questions were circulated to Toronto’s Drug Checking Service collection site member organizations—Parkdale Queen West Community Health Centre, South Riverdale Community Health Centre, The Works at Toronto Public Health, and Moss Park Consumption and Treatment Service—for input. Since then, this resource has been shared with the Canadian Centre on Substance Use and Addiction’s National Drug Checking Working Group, the Alliance for Collaborative Drug Checking, and many other community-based organizations.

From the initial draft in April 2022, this resource has undergone two iterations and is now on version three. It is anticipated that iterative improvements to this document will result in a number of future versions being released. As the document states, “We are learning too, but are a resource to the community and could attempt to assist with translation if that would be helpful to you. You can reach us at drugchecking@cdpe.org”. The *Onsite drug checking technology purchase and partnership considerations* questions are not meant to be exhaustive but provide a foundation or place to start for community organizations communicating with for-profit drug checking technology vendors. Questions about acceptable vendor responses and requests to collaborate on improvements or adaptations of these questions are welcomed by Toronto’s Drug Checking Service.

## 4. Conclusions

In New Zealand, drug checking providers must choose from a select number of technologies and testing methods that have been approved by the Ministry of Health’s drug checking licensing team [13]. For countries such as Canada, where technologies are not required to be evaluated before they go to market, Toronto’s Drug Checking Service believes it is paramount that organizations are equipped to make the most informed decision possible when investing in drug checking technologies; the *Onsite drug checking technologies purchase and partnership considerations* resource aims to meet this need [12].

## 5. Future Directions

Technologies created exclusively for the purposes of drug checking are emerging in North America, Europe, and likely elsewhere. Ideally, these technologies will offer improvements to those currently available, providing people who use drugs with comprehensive qualitative and quantitative information for all drugs and non-drug fillers found in their sample in a matter of minutes for a reasonable price. To date, no single technology used for drug checking meets these criteria and, therefore, remaining skeptical and curious about the limitations of available technologies is critical for community agencies looking to offer drug checking services.

## Data Availability

Data presented in this publication is openly available: https://drugchecking.cdpe.org/.

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
