# Peer review of "Considerations for Purchasing Drug Checking Technologies: Perspectives from Toronto’s Drug Checking Service"

_ijerph, 2023, doi:10.3390/ijerph20156486_

Round 1

Reviewer 1 Report

Thanks for the opportunity of assessing this manuscript.

Personally, I am a big fan of practictioners' engagement in academic research, as we do need to bridge the gap between theory and practice. So, I think that the Toronto’s Drug Checking Service, Centre on Drug Policy Evaluation, St. Michael’s Hospital's contribution may be absolutely relevant. 

However, the paper in its current version is too skinny. Therefore, I would suggest the authors to improve it, especially the introduction and practical implications. 

Also the reference list is too poor. I would recommend the authors to read some literature reviews like this one (which was recently published by IJERPH:

Biancuzzi H et al. (2022) Opioid Misuse: A Review of the Main Issues, Challenges, and Strategies, International Journal of Environmental Research and Public Health, Vol. 19, issue 18, article no. 11754. https://doi.org/10.3390/ijerph191811754

This paper (as well as others) may help you to better frame the topic and find a more rigorous academic background. 

Good luck with the development of your paper. 

Reviewer 2 Report

The manuscript by Thompson and McDonald describes the different uses of drug checking technologies based on the knowledge acquired from the services provided by Toronto's Drug Checking Service from October 2019 to November 2022. The manuscript also provides a list of questions to make informed decisions when it comes to purchasing drug checking technologies. The manuscript is well-written and described for organizations interested in developing and operating a drug checking service. However, additional information is suggested:

1. In regards to security, how are the facility and personnel secured? What are the hours of operation?

2. If the drug is contaminated, what is the following procedure? Is the rest of the drug safely discarded?

Reviewer 3 Report

This manuscript promises to be a welcome addition to the literature as it gives concrete, experience-based advice on considerations for community-based organizations to select drug checking equipment. The list of questions is thoughtful, comprehensive, and well-organized, and will likely help new services avoid pitfalls and launch more quickly, and in turn help the field of drug checking expand and deliver services more effectively.

My main comment is that while the list of questions is valuable, it may be of limited use to those who are not positioned to evaluate the vendor's answers. That is, it would be helpful to also include some guidance on what to look out for in an answer, what separates an acceptable answer from an unacceptable one, any gold standards, etc. Alternatively, the list could also include guidance on what partnerships and expertise need to be in place in order to assess answers in each section.

Another general comment is that it would be helpful to know more about the process of how the list was developed, i.e., who was consulted, how were decisions made about questions to include/exclude, etc. This might for example help organizations who want to adapt the list or develop something similar.

Specific comments:

- Lines 32-33 state that TDCS was the first and only drug checking service in Ontario, but other services such as Sandy Hill in Ottawa (see https://www.shchc.ca/programs/oasis/drug-checking) and event/nightlife-based services such as TRIP in Toronto were operating before 2019. Could the authors clarify/specify what sets TDCS apart?

- Information on what technologies TDCS uses and why they were selected would be a better fit in the overview of TDCS (starting on line 44). The information that drug checking technologies are not regulated in Canada (lines 169-174) would also be a better fit in the introduction to set the stage for the importance of the list.

- Lines 146-147: The examples given for this question are not clear, please clarify the information in parentheses following the question.

Additional suggestions (for consideration only):

- The authors are generally very careful and measured with their statements, but the statement on lines 70-73, that FTIR + test strips should be adequate for non-opioids, could be risky as contamination of different types of substances varies by region. The authors could consider qualifying the statement further.

- Language: Some people prefer using "unintentional drug poisoning" to "drug overdose" (lines 26, 31, 197) as the latter can be perceived as stigmatizing and the former points more explicitly to the unpredictable drug supply as a source of the problem. The authors could consider replacing these terms.

Round 2

Reviewer 1 Report

Thanks for amending the manuscript.

I think that the current version is better. Still, reporting a practical experience is not an excuse not to try to be rigorous. Therefore, unless the authors provide a good list of references to frame their research, I will not feel comfortable in recommending it for publication. 

Author Response

Hi Reviewer 1, thank you for taking the time to provide feedback on our revised manuscript. At this time, we will not be pursuing further edits to our Opinion piece. As two non-researchers, with no bandwidth for publications like this, we feel we've tried our best to contextualize and support our Opinion piece with references (adding four references we felt framed our opinion). By no means are we trying to use this as "an excuse not to try to be rigorous", but we felt we did have a good list of references - I suppose this is a difference of opinion. Regardless, this has been a good learning experience. If in the future you felt there was an opinion piece that was adequately referenced we'd appreciate you sending it along to us so we understand what it would have taken to meet your standard - [email protected]. Thanks!